# AURA: Augmented Representation for Unified Accuracy-aware Quantization

## Abstract

Low-bit quantization is essential for the efficient deployment of Large Language Models (LLMs), but it often degrades model accuracy due to activation outliers concentrated in a few critical channels. Current solutions face a trade-off: Transformation-based methods generate significant additional overhead due to offline parameter learning and online execution; while efficient mixed-precision schemes, despite their high efficiency, lack forward compatibility because they rely on customized General Matrix Multiplication (GEMM) kernels. To address this dilemma, we introduce AURA (Augmented Representation for Unified Accuracy-aware Quantization). AURA employs a theoretically-grounded, accuracy-aware metric to identify critical channels and compensates for their quantization errors by constructing a unified, low-bit augmented matrix. This design decouples the error compensation logic from the underlying GEMM kernel, enabling high performance on current hardware while ensuring robust adaptability to future architectures. Our experiments on Llama and Qwen models demonstrate that AURA achieves state-of-the-art results for 4-bit quantization across a wide range of benchmarks. In system performance, our framework significantly outperforms TensorRT-FP8 baselines, achieving a nearly 3-fold reduction in prefill latency and reducing peak decoding memory to one-third of the FP16 baseline.

## 1 Introduction

Large Language Models (LLMs) have emerged as a foundational technology, with remarkable capabilities that are reshaping scientific and industrial applications (Vaswani et al., 2023; Brown et al., 2020). This exceptional performance, however, comes at the cost of prohibitive computational and memory requirements, presenting a significant barrier to their widespread adoption on resource-constrained devices. Quantization, particularly to sub-byte precisions like 4-bit formats, is a key strategy for mitigating these demands, offering drastic reductions in resource footprint while enabling significant acceleration on specialized hardware (Yao et al., 2022; Xiao et al., 2024).

The primary obstacle in sub-byte quantization is managing extreme activation outliers, which contrast sharply with the well-behaved distributions of model weights (Dettmers et al., 2022; Xiao et al., 2024). This challenge has led to weight-only quantization methods, such as GPTQ (Frantar et al., 2023) and AWQ (Lin et al., 2024), which circumvent this issue by leaving activations in full precision. While effective for model compression, this approach fails to accelerate computation or reduce activation-related memory bandwidth. To achieve complete end-to-end acceleration, quantization of both weights and activations is necessary. However, standard affine schemes for quantizing both are ill-equipped to handle such outliers, forcing a difficult trade-off between clipping vital information and degrading precision for other values. Seminal works have established that these outliers are concentrated within a small subset of "critical" channels (Dettmers et al., 2022; Heo et al., 2023). Preserving information within these critical channels thus emerges as the central challenge for modern Post-Training Quantization (PTQ).

To address this challenge, two dominant paradigms have emerged. They aim to preserve information by either reshaping activation distributions to mitigate outliers or by selectively allocating higher precision to critical channels. The first, pre-quantization transformation (Ma et al., 2024; Liu et al., 2024; Shao et al., 2024), offers generality but incurs substantial overhead from parameter learning and runtime execution. The second, mixed-precision quantization (Zhao et al., 2024;

Ashkboos et al., 2023), achieves high performance through bespoke kernels for heterogeneous data formats. This approach, however, creates a tight coupling between the algorithm and specific hardware, severely limiting its portability and forward-compatibility. As a result, the field faces a trade-off between quantization effectiveness, computational overhead, and architectural adaptability.

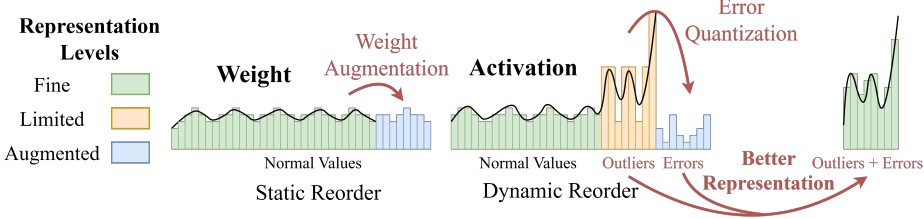

Figure 1: Overview of AURA

To address this trade-off, we introduce AURA (Augmented Representation for Unified Accuracy-aware Quantization), a novel framework that operates on a "compensate-and-fuse" paradigm. AURA employs a theoretically-grounded metric to precisely identify critical channels and efficiently compensates for their quantization errors. Its core mechanism is the construction of a unified low-bit augmented matrix that fuses the main matrix multiplication with its precision compensation into a single, standard GEMM operation.

We validated AURA experimentally across a wide range of benchmarks and found that it consistently surpasses or matches state-of-the-art quantization methods. In evaluations covering 0-shot accuracy, perplexity, math, and code generation, this high fidelity is achieved with remarkable efficiency: compensating for just 6% of critical channels allows AURA to recover nearly 54% of the accuracy gap relative to the FP16 baseline. System-level evaluations further demonstrate a 76% prefill latency speedup over the TensorRT-FP8 baseline, while reducing peak decoding memory to one-third of the FP16 requirement. These results are underpinned by three core contributions:

**Decoupled Quantization Framework:** AURA's core design principle is the decoupling of the quantization algorithm from the GEMM kernel, a choice crucial for engineering simplicity and maintainability. Unlike mixed-precision approaches that require custom GEMM kernels, AURA's logic is fully contained within the quantization stage. This modularity proves both effective and adaptable: the framework consistently enhances the performance of diverse data formats (e.g. MXFP4, INT4) simply by adjusting the quantization kernel, leaving the underlying, highly-optimized GEMM implementation untouched (see Appendix D.5).

**Robust Critical Channel Identification:** We propose a theoretically-grounded, accuracy-aware sensitivity metric. The metric's generality is demonstrated by strong performance across diverse tasks, including 0-shot, math, and code, using only a single WikiText2 calibration set. Its robustness is further validated by its ability to maintain stable accuracy even when the calibration domain is shifted to different corpora, as shown in Table 4.

**High-Performance Fused Kernel Design:** We developed a high-performance inference pipeline built on kernel fusion. At its core is a single GEMM call, which is significantly more efficient than executing separate computation and compensation kernels (Appendix D.2). We also fuse the complex pre-processing steps into an optimized CUDA kernel. These steps include channel reordering, quantization, error quantization, and RMSNorm. As illustrated in Figure 6, this fused design delivers a consistent latency advantage over the TensorRT-FP8 baseline across all compensation ratios.

## 2 PRELIMINARY

The core operation in a transformer's linear layer is $\boldsymbol{Y} = \boldsymbol{X}\boldsymbol{W}^T$. Post-training quantization (PTQ) approximates this operation as $\boldsymbol{Y}_q = Q(\boldsymbol{X})Q(\boldsymbol{W})^T$, where $Q(\cdot)$ is a low-bit quantization function. The resulting output error, $\boldsymbol{E}_Y = \boldsymbol{Y} - \boldsymbol{Y}_q$, can be decomposed into three primary components:

$$\boldsymbol{E}_Y = \underbrace{\boldsymbol{E}_X Q(\boldsymbol{W})^T}_{\text{Term A: Activation Error}} + \underbrace{Q(\boldsymbol{X})\boldsymbol{E}_W^T}_{\text{Term B: Weight Error}} + \underbrace{\boldsymbol{E}_X \boldsymbol{E}_W^T}_{\text{Term C: Second-Order Error}} \quad (1)$$

It is widely established that Term A, the activation error, is the primary source of accuracy degradation in LLMs. This principle is empirically validated by the success of weight-only quantization methods, such as GPTQ (Frantar et al., 2023) and AWQ (Lin et al., 2024), which keep activations in full precision. This approach effectively nullifies the activation error matrix ($\boldsymbol{E}_X = 0$), thereby eliminating the dominant error sources (Term A and Term C) and achieving near-lossless compression. The reason for this disparity is that weight distributions are typically well-behaved and Gaussian-like, leading to small errors ($\boldsymbol{E}_W$), whereas activation distributions exhibit extreme outliers that create a large, structured error matrix ($\boldsymbol{E}_X$). A detailed derivation is provided in Appendix B.

To mitigate activation-induced error in full quantization, modern methods rely on fine-grained techniques like group-wise quantization and native block-scaling formats (e.g., NVFP4 and MXFP). These methods partition a tensor into small groups, each with its own scaling factor, allowing the quantizer to adapt to localized data distributions. Although highly effective, the fixed granularity of these formats is a compromise; a single outlier can still degrade the precision for all other channels within its block. This creates a significant residual error that, while smaller, remains the key obstacle to achieving near-lossless performance. AURA is designed to directly address and compensate for this residual error. Further details on these formats are provided in Appendix C.

## 3 MOTIVATIONS

**Motivation 1: Accuracy-Aware Identification of Critical Channels**

Accurate identification of critical channels is a prerequisite for effective quantization, yet existing paradigms have notable limitations. Rotation-based methods (Ashkboos et al., 2024) apply an indiscriminate and holistic transformation, lacking the precision to target the true sources of error. The more common magnitude-based heuristic ($|\boldsymbol{X}|$) is also flawed; it is theoretically incomplete because it ignores the modulating effect of weight norms. It is also practically sensitive to the statistical limitations of a finite calibration set, often failing to capture the input-dependent variance of outliers. These limitations highlight the need for a targeted, theoretically-grounded metric that is robust to distributional shifts and can accurately identify the primary sources of quantization error.

**Motivation 2: Low-bit Compensation with Unified Representation**

Efficiently compensating for quantization errors is a key challenge. Mixed-precision schemes (Zhao et al., 2024; Ashkboos et al., 2023) address this by retaining critical channels in higher precision, but this approach creates heterogeneous data formats that require complex, customized kernels. We propose an alternative: we hypothesize that fine-grained formats like NVFP4 are sufficiently expressive to model their own primary quantization error ($\boldsymbol{E}_X$). This insight leads to our core mechanism: a unified low-bit augmented representation that concatenates the primary data with the quantized error of critical channels, as shown in Equation 2. The subscripts "n" and "c" denote non-critical and critical channels, respectively.

$$\boldsymbol{X}_{aug} = [Q(\boldsymbol{X}_n) \,|\, Q(\boldsymbol{X}_c) \,|\, Q(\boldsymbol{E}_c)], \boldsymbol{W}_{aug} = [Q(\boldsymbol{W}_n) \,|\, Q(\boldsymbol{W}_c) \,|\, Q(\boldsymbol{W}_c)] \qquad (2)$$

The efficacy of this low-bit compensation is visualized in Figure 2. The figure demonstrates that AURA sharply suppresses the error norms of critical channels, validating that a unified low-bit representation is an effective method for achieving high-fidelity quantization without the burden of mixed precision.

**Motivation 3: Decoupling Quantization from GEMM Execution**

The augmented matrix formulation shown in Equation 2 is the cornerstone of our third principle: decoupling the quantization algorithm from the underlying GEMM execution. By combining the main computation and its error correction into a monolithic matrix, the entire operation is reduced to a single standard GEMM call. This design elegantly sidesteps the engineering fragility of mixed-precision methods, which require burdensome bespoke kernels. Consequently, AURA eliminates maintenance overhead and can readily leverage performance gains from optimized libraries such as CUTLASS. This ensures a robust and forward-compatible engineering solution for current and future hardware (see Appendix D.5 for details).

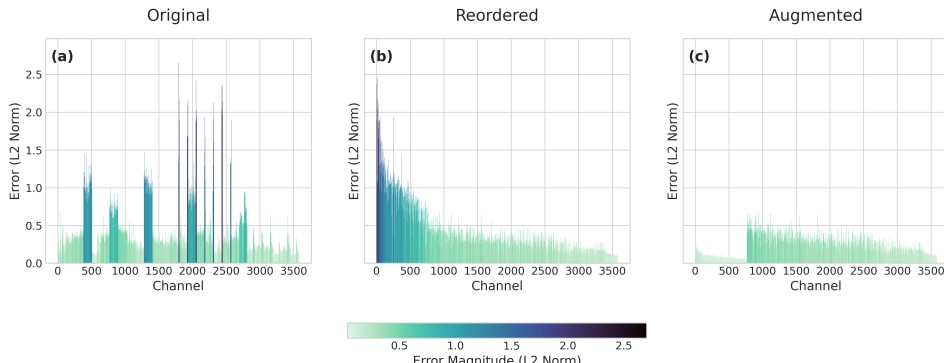

Figure 2: Efficacy of low-bit error compensation in an o_proj layer of Qwen2.5-7B. The y-axis represents the L2 norm of the NVFP4 quantization error. (a) Error norms for channels in their original order. (b) Channels reordered by sensitivity, concentrating the largest errors. (c) Residual error norms after compensation, showing a dramatic suppression of the error spikes.

## 4 METHOD

### 4.1 ACCURACY-AWARE SENSITIVITY METRIC

To accurately identify the sources of quantization error, we developed a novel sensitivity metric that considers the interaction between activation errors and weights. The core idea is to decompose the total output error matrix into a sum of outer products, where each term represents the contribution of a single input channel. This method allows us to quantify the impact of each channel on the final error and to identify those most affected by quantization. This section details the theoretical foundation and implementation of our approach.

#### 4.1.1 THEORETICAL FOUNDATION

As established in our preliminary analysis, the dominant output error component is $\boldsymbol{E}_Y \approx \boldsymbol{E}_X \boldsymbol{W}^T$. We define this error matrix, $\boldsymbol{E}$, as a sum of rank-1 matrices, where each matrix represents the error contribution from a single input channel:

$$\boldsymbol{E} = \boldsymbol{E}_X \boldsymbol{W}^T = \sum_{c=1}^{C_{in}} \boldsymbol{e}_c \boldsymbol{w}_c^T \tag{3}$$

where $\boldsymbol{e}_c$ and $\boldsymbol{w}_c$ are the column vectors of the activation error matrix $\boldsymbol{E}_X$ and the weight matrix $\boldsymbol{W}$ for a given channel $c$, and $C_{in}$ is the total number of input channels.

To quantify these contributions, we employ the Frobenius norm ($\|\cdot\|_F$, where $\|\boldsymbol{A}\|_F = (\sum_{i,j} a_{ij}^2)^{1/2}$). The triangle inequality for matrix norms provides a theoretical motivation for our approach, as it bounds the total error norm by the sum of individual channel-wise error norms:

$$\|\boldsymbol{E}\|_F = \left\| \sum_{c=1}^{C_{in}} \boldsymbol{e}_c \boldsymbol{w}_c^T \right\|_F \leq \sum_{c=1}^{C_{in}} \|\boldsymbol{e}_c \boldsymbol{w}_c^T\|_F \tag{4}$$

This inequality suggests that the total error is dominated by channels with the highest contribution norms. We therefore define the sensitivity $S_c$ of a channel $c$ as the norm of its contribution term:

$$S_c = \|\boldsymbol{e}_c \boldsymbol{w}_c^T\|_F \tag{5}$$

This score provides a principled ranking to prioritize compensation for the most critical channels.

#### 4.1.2 IMPLEMENTATION

A direct computation of $S_c$ by forming $C_{in}$ large outer-product matrices would be highly inefficient. Fortunately, the Frobenius norm of a rank-1 matrix is mathematically equivalent to the product of

the L2 norms of its constituent vectors. This identity allows for an efficient and exact calculation:

$$S_c = \|\boldsymbol{e}_c \boldsymbol{w}_c^T\|_F = \|\boldsymbol{e}_c\|_2 \cdot \|\boldsymbol{w}_c\|_2 \tag{6}$$

This identity transforms the problem into a computationally tractable task. Instead of manipulating large matrices, we only need to compute vector norms, which is a low-cost operation.

In practice, during the offline calibration phase, we first compute the activation error matrix $\boldsymbol{E}_X$ for the entire calibration set. Then, for each channel $c$ in every linear layer, we calculate its sensitivity score $S_c$ using Equation 6. These scores are then ranked to determine the precise set of critical channels to be compensated for during inference. This entire process is computationally inexpensive.

## 4.2 KERNEL DESIGN

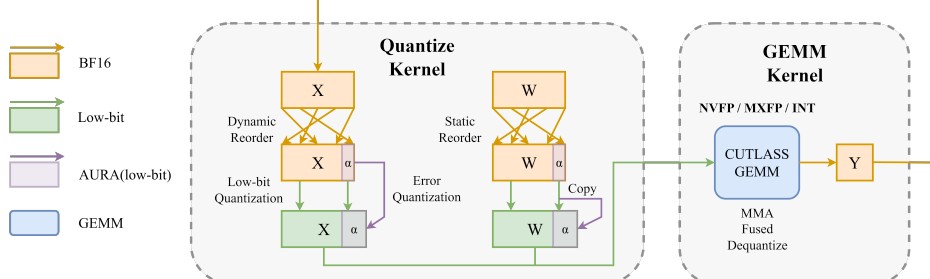

Figure 3: Kernel Design of AURA

Building on the sensitivity metric, the primary role of our kernel design is to implement the augmented matrix formulation introduced in our motivations. It efficiently constructs the augmented tensors ($\boldsymbol{X}_{aug}$ and $\boldsymbol{W}_{aug}$), enabling the entire computation to be performed in a single, standard matrix multiplication ($\boldsymbol{X}_{aug} \cdot \boldsymbol{W}_{aug}^T$). This is achieved through a decoupled architecture consisting of two key components: a fused quantization kernel and a unified GEMM kernel.

### 4.2.1 QUANTIZATION KERNEL

To efficiently manage sparse outliers and ensure contiguous memory access for their compensation, our quantization kernel first adopts a channel reordering strategy similar to Atom and RPTQ (Zhao et al., 2024; Yuan et al., 2023). Based on pre-computed sensitivity scores from the calibration set, critical channels are permuted to the end of the channel dimension. Consequently, the weight matrix $\boldsymbol{W}$ is statically reordered offline, while the activation matrix $\boldsymbol{X}$ is dynamically reordered during inference; the identical permutation ensures the correctness of the subsequent matrix multiplication.

After this reordering, the kernel processes the activation matrix $\boldsymbol{X}$. Non-critical channels undergo standard low-bit quantization, while for critical channels, the kernel also computes the residual error ($\boldsymbol{E}_X$) and quantizes this error term into the same low-bit format for augmentation. The weight matrix $\boldsymbol{W}$ is prepared analogously, where the weights of the critical channels are duplicated to populate the augmented section. This entire process, including the preceding RMSNorm operation where applicable, is fused into a single CUDA kernel to minimize memory I/O and latency.

### 4.2.2 GEMM KERNEL

The augmented matrix formulation is implemented using a single unified GEMM kernel built with the CUTLASS library. This kernel accepts the monolithic low-bit augmented tensors ($\boldsymbol{X}_{aug}$, $\boldsymbol{W}_{aug}$) and produces a higher precision output, typically BFloat16. The kernel is agnostic to the compensation ratio, which simply manifests as a larger K-dimension in the matrix multiplication. This unified approach is critical for performance, as it eliminates the overhead of an additional kernel launch required by a bifurcated strategy that would perform two separate GEMM calls (a detailed performance comparison is provided in Appendix D.2).

Furthermore, for block-scaling formats like NVFP4 and MXFP, our kernel leverages the latest hardware innovations. On modern architectures such as Blackwell, it utilizes new Tensor Core instruc-

tions (e.g. `tcgen05.mma`) that provide native hardware support for dequantization (NVIDIA Corporation, 2024b). This allows the low-bit format to be expanded directly to the accumulation precision on high-performance Tensor Cores. This design circumvents the dequantization bottleneck of slower CUDA cores and ensures maximum hardware efficiency (Lin et al., 2025).

### 4.2.3 INTEGRATION INTO THE DECODER LAYER

We integrate AURA by replacing each `nn.Linear` module with our equivalent, which consists of a custom quantization kernel and a unified GEMM kernel. To maximize performance, this quantization process, including activation reordering, is fused with the preceding RMSNorm operation. This fusion eliminates an intermediate memory pass, thereby reducing I/O latency. Overall decoder performance is further enhanced by leveraging the FlashInfer (Ye et al., 2024) library for efficient INT4 KV Cache and attention computations.

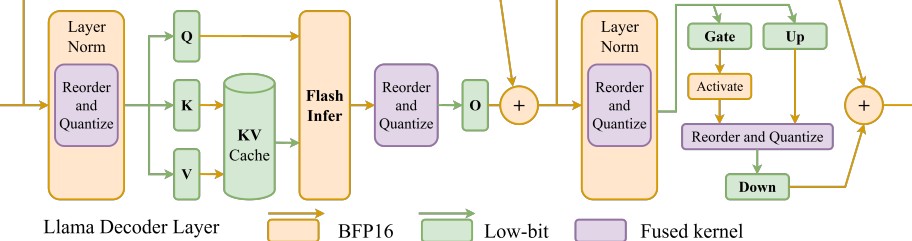

Figure 4: Integrating AURA to Transformer Blocks in LLM

## 5 EXPERIMENTS

### 5.1 EXPERIMENTAL SETUP

**Quantization:** To demonstrate AURA's compatibility with advanced data formats, we implement our framework by quantizing both activations and weights to the NVFP4 format. We use a calibration set consisting of 32 samples from the WikiText2 (Merity et al., 2016) dataset, each with a sequence length of 2048, to determine the channel sensitivity scores and the corresponding reordering permutation. The compensation ratio is a tunable hyperparameter that balances accuracy and performance. As detailed in Appendix D.1, our analysis indicates that a compensation ratio in the 4-6% range provides an optimal trade-off. Consequently, we use this range for our main results. Table 1 summarizes additional implementation details.

Table 1: Compensation ratio, quantized size, and time cost for AURA across different models.

| Model | Ratio | Model Size (GB) | Calibration Time (s) | Quantization Time (s) |
|-------|-------|-----------------|----------------------|------------------------|
| Llama3.1-8B | 4% | 4.32 | 100 | 11 |
| Qwen2.5-7B | 6% | 4.16 | 86 | 11 |
| Qwen2.5-32B | 6% | 18.71 | 270 | 50 |

**Baselines and Benchmarks:** To ensure a comprehensive comparison, we benchmark AURA against a diverse suite of state-of-the-art methods on the Llama (Grattafiori et al., 2024) and Qwen (Qwen et al., 2025) models. Our baselines cover key strategies spanning both integer and floating-point paradigms: transformation-based methods (FlatQuant, QuaRot) (Sun et al., 2024; Ashkboos et al., 2024), mixed-precision quantization (Atom) (Zhao et al., 2024), and hardware-native format approaches (NVFP4) (NVIDIA Corporation, 2024a). For evaluation, we use a range of benchmarks. Zero-shot performance is measured using ARC-C (Clark et al., 2018), Hellaswag (Zellers et al., 2019), Winogrande (Sakaguchi et al., 2019), BoolQ (Clark et al., 2019), and PIQA (Lourie et al., 2021). We evaluate perplexity (PPL) on the WikiText2 (Merity et al., 2016) dataset. Additionally, we benchmark the Qwen2.5 series on domain-specific tasks, using HumanEval (Chen et al., 2021a)

and Mbpp (Austin et al., 2021) for code generation. Implementation details for the math benchmarks are provided in Appendix D.6.

## 5.2 MAIN RESULTS.

Table 2: Comparison of zero-shot accuracy (↑, higher is better) and WikiText2 PPL (↓, lower is better) for different quantization methods. Our method, AURA, is shown in bold. We report results on ARC-C, BoolQ (BQ), Hellaswag (HG), PIQA (PQ), and Winogrande (WG) benchmarks.

| Model | Method | 0-shot | | | | | | PPL |
|---|---|---|---|---|---|---|---|---|
| | | ARC-C | BQ | HG | PQ | WG | Avg. | |
| Llama3.1-8B | FP16 | 53.58 | 81.99 | 78.90 | 81.18 | 74.03 | 73.94 | 6.24 |
| | FlatQuant | **50.00** | 78.99 | 76.80 | 79.16 | 72.69 | 71.53 | **6.98** |
| | QuaRot | 45.05 | 75.93 | 73.51 | 77.20 | 68.35 | 68.01 | 7.83 |
| | Atom | 47.53 | 77.80 | 74.22 | 78.02 | 69.46 | 69.41 | 7.52 |
| | NVFP4 | 49.06 | 73.94 | 77.16 | 79.76 | **72.77** | 70.54 | 7.17 |
| | **AURA** | 49.83 | **79.20** | **77.41** | **79.98** | 72.38 | **71.76** | 7.07 |
| Qwen2.5-7B | FP16 | 51.28 | 84.68 | 78.98 | 79.71 | 73.24 | 73.58 | 6.85 |
| | FlatQuant | 49.40 | 83.52 | 76.61 | 79.33 | 68.59 | 71.49 | 7.87 |
| | QuaRot | 48.46 | 81.25 | 75.90 | 78.40 | 67.48 | 70.30 | 7.74 |
| | Atom | 48.38 | 81.56 | 74.63 | 77.64 | 68.59 | 70.16 | 8.96 |
| | NVFP4 | 50.77 | 82.84 | 77.29 | 78.73 | 69.14 | 71.75 | 7.43 |
| | **AURA** | **51.02** | **83.91** | **77.45** | **79.87** | **71.43** | **72.74** | **7.34** |
| Qwen2.5-32B | FP16 | 55.72 | 87.58 | 84.14 | 82.10 | 75.45 | 77.00 | 5.02 |
| | FlatQuant | 55.80 | **86.64** | 82.79 | **82.21** | 76.32 | 76.75 | 5.46 |
| | QuaRot | 51.37 | 85.29 | 82.07 | 80.52 | 72.14 | 74.28 | 5.90 |
| | Atom | 54.10 | 86.42 | 82.38 | 80.30 | 73.40 | 75.32 | 5.83 |
| | NVFP4 | 55.12 | 85.38 | **83.28** | 81.61 | 75.61 | 76.20 | 5.42 |
| | **AURA** | **56.57** | 86.48 | 83.05 | 81.77 | **77.11** | **77.00** | **5.40** |

Table 2 compares AURA against selected baselines on zero-shot downstream tasks and WikiText2 (Merity et al., 2016) perplexity. A key finding is the efficiency of our accuracy-aware compensation strategy: by augmenting just 6% of critical channels, AURA recovers nearly 54% of the accuracy gap between naive NVFP4 quantization and the FP16 baseline.

This efficiency leads to state-of-the-art average accuracy on general benchmarks. The framework's performance is highly comparable to the FP16 baseline: on Llama3.1-8B, AURA retains over 97% of the original performance, while on Qwen2.5-7B, it performs within 0.9 percentage points of the FP16 baseline. This strong performance extends to specialized domains that require precise reasoning. As shown in Table 3, AURA preserves the code generation capabilities of the Qwen-Coder model, with performance nearly identical to FP16 and superior to that of Atom. Similarly, AURA achieves strong results on math reasoning tasks (see Appendix D.6). Collectively, these results establish AURA as a robust solution for achieving high-accuracy, low-bit quantization across diverse model types and tasks.

Table 3: Performance on code generation benchmarks for the Qwen2.5-Coder-7B-Instruct model. We report pass@1 accuracy.

| Method | HumanEval | HumanEval+ | MBPP | MBPP+ |
|---|---|---|---|---|
| FP16 | 84.1 | 79.9 | 76.9 | 65.2 |
| Atom | 80.5 | 76.2 | 74.5 | 63.2 |
| **AURA** | **84.1** | **79.9** | **75.7** | **64.2** |

## 5.3 ABLATION STUDIES

### 5.3.1 SELECTION STRATEGIES

We validate our proposed accuracy-aware sensitivity metric by comparing its performance against a conventional magnitude-based heuristic. This baseline selects channels based on their mean absolute activation values, which are computed on the WikiText2 (Merity et al., 2016) calibration set. The evaluation is based on the average zero-shot accuracy across several benchmarks: ARC-C (Clark et al., 2018), BoolQ (Clark et al., 2019), PIQA (Lourie et al., 2021), Lambada (Paperno et al., 2016), and Winogrande (Sakaguchi et al., 2019). Figure 5 illustrates the trade-off between model accuracy and the percentage of compensated channels, focusing on the practical 0-10% range. Our accuracy-aware metric consistently outperforms the magnitude-based baseline, demonstrating its superior ability to identify the channels most critical to model performance.

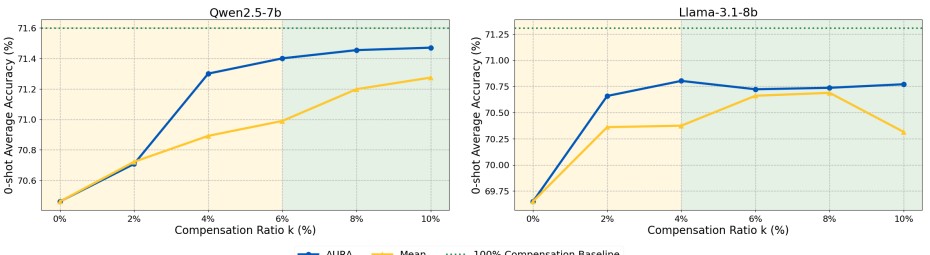

Figure 5: Selection strategies.

### 5.3.2 CALIBRATION DATASETS

To evaluate AURA's robustness to the choice of calibration data, we derive its quantization parameters using three distinct datasets: WikiText2 (Merity et al., 2016) (encyclopedic text), C4 (Raffel et al., 2019) (web text), and HumanEval (Chen et al., 2021b) (code). Table 4 reports the performance of the resulting Llama2-7B models. The results demonstrate remarkable stability: the average zero-shot accuracy remains consistent, and perplexity also shows minimal variation.

Table 4: The effect of different calibration datasets on the performance of AURA on Llama2-7B. We report 0-shot accuracy (↑) and WikiText2 perplexity (PPL) (↓).

| Method | Calibration | 0-shot Accuracy | | | | | | PPL |
|--------|-------------|------|------|------|------|------|------|------|
| | | ARC-C | BQ | HG | PQ | WG | Avg. | |
| FP16 | — | 46.25 | 77.74 | 75.92 | 79.00 | 68.98 | 69.58 | 5.47 |
| **AURA** | WikiText-2 | 44.03 | 76.27 | 74.84 | 78.18 | 68.27 | 68.32 | 5.78 |
| | C4 | 43.52 | 76.76 | 74.86 | 78.35 | 68.75 | 68.45 | 5.77 |
| | Human-Eval | 43.77 | 76.36 | 74.55 | 78.29 | 68.59 | 68.31 | 5.78 |

## 5.4 EFFICIENCY EVALUATION

### 5.4.1 PERFORMANCE TRADE-OFFS OF THE COMPENSATION RATIO

To quantify the performance implications of our framework, we ablated the compensation ratio and evaluated its impact on prefill latency and peak memory. As illustrated in Figure 6, our analysis reveals a clear sub-linear relationship between the compensation ratio and the resulting performance overhead. Notably, doubling the theoretical GEMM workload with a 100% compensation ratio increases prefill latency by only 54% and peak memory by 45%. This validates our choice of a small ratio for practical deployments, as this approach provides substantial accuracy gains for a marginal performance overhead. Crucially, our AURA implementation remains faster than the TensorRT-FP8 baseline across all configurations, underscoring the efficiency of our unified kernel design.

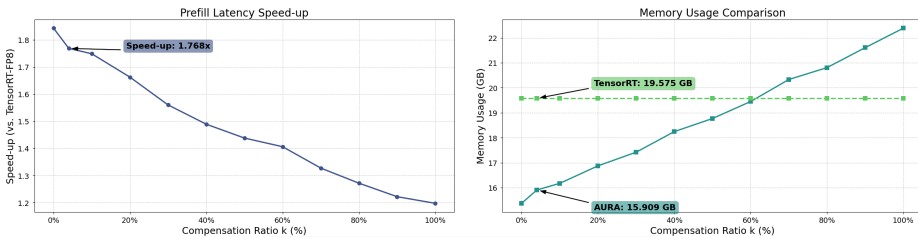

Figure 6: Performance Trade-offs of Compensation Ratio on Llama2-13B. The end-to-end prefill speedup over the TensorRT-FP8 (per-tensor) baseline and peak memory usage were measured for a range of compensation ratios from 0% to 100%. Experiments were conducted on an NVIDIA RTX 5090 with a batch size of 32 and a sequence length of 2048.

### 5.4.2 END-TO-END PERFORMANCE

To evaluate AURA's practical system performance, we benchmarked its end-to-end inference on the Llama2-7B and 13B models. We measured prefill latency and peak memory usage during the decoding phase. We conducted these experiments on an RTX 5090 across various sequence lengths, with batch sizes fixed at 8 for the 7B model and 2 for the 13B model due to memory constraints. As shown in Figure 7, the results demonstrate substantial improvements over the FP16 baseline. AURA achieves a nearly 3-fold reduction in prefill latency. In the decoding phase, it reduces peak memory usage to less than one-third of the FP16 baseline. These findings validate AURA as a highly efficient solution for practical LLM serving.

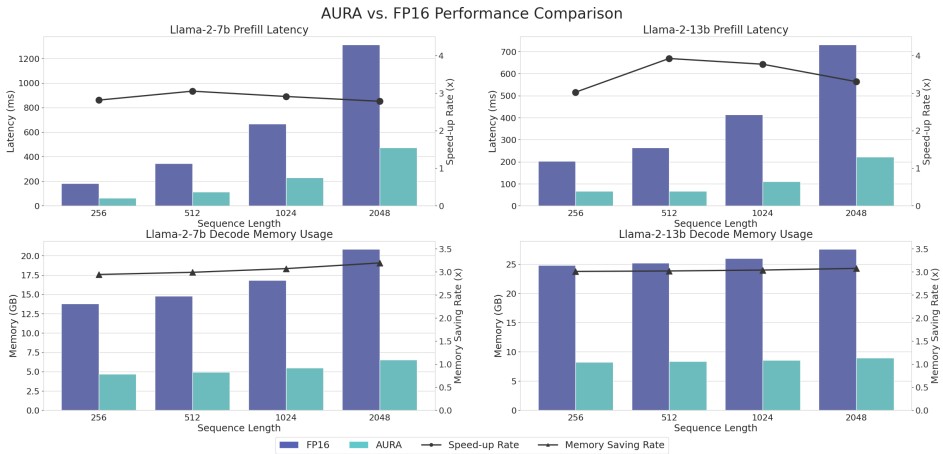

Figure 7: End-to-end efficiency for AURA

## 6 CONCLUSIONS

We introduced AURA, a quantization enhancement framework that addresses the trade-off between accuracy and performance in low-bit quantization. Its core mechanism, an augmented matrix, fuses error compensation for critical channels into a single standard GEMM operation, thus decoupling the quantization algorithm from hardware execution. Experimentally, AURA achieves state-of-the-art accuracy, recovering nearly 54% of the FP16 accuracy gap with only 6% channel compensation. In terms of performance, our implementation demonstrates a consistent speedup over the TensorRT-FP8 baseline while also reducing prefill latency and peak memory usage in end-to-end inference. AURA thus provides a practical and forward-compatible solution for efficient LLM deployment. Future work will explore dynamic compensation and sub-4-bit regimes.

## REPRODUCIBILITY STATEMENT

We are committed to ensuring that all results presented in this paper are reproducible. To facilitate this, we provide the following details regarding our data, code, and experimental setup:

**Data Availability:** All datasets used for calibration and evaluation are publicly available and are standard benchmarks in the field. The primary calibration dataset, WikiText2, as well as all downstream benchmark datasets (e.g. ARC, Hellaswag, HumanEval), can be accessed through `https://huggingface.co/`.

**Code Availability:** The source code for the AURA framework, including the implementation of our custom fused CUDA kernels and the scripts required to reproduce all experiments, is provided in `https://anonymous.4open.science/r/AURA-1941/`.

**Experimental Setup:** A detailed description of our experimental setup, including the specific model versions (Llama, Qwen), hardware (NVIDIA RTX 5090), software libraries, and key hyperparameters (e.g. compensation ratios, batch sizes, sequence lengths), is provided in Section 5 .

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

## A  RELATED WORKS

**Mixed-precision quantization** aims to balance accuracy and efficiency by allocating higher precision to sensitive components, such as outlier activation channels, while quantizing the rest to a lower bit-width (Dettmers et al., 2022; Saxena et al., 2025; Ashkboos et al., 2023; Hooper et al., 2025). A notable example is Atom (Zhao et al., 2024), which preserves 128 critical channels in INT8 and the remainder in INT4, achieving high performance on specific hardware like the RTX 4090. However, this approach often suffers from poor portability and hardware-unfriendliness, as it typically requires bespoke GEMM kernels to handle the heterogeneous data formats.

AURA is inspired by the principle of selectively addressing outlier channels but refines it with a more theoretically-grounded sensitivity metric. Furthermore, instead of relying on high-precision formats, AURA compensates for errors using a unified low-bit augmented matrix, thereby circumventing the need for custom kernels and ensuring greater hardware compatibility.

**Pre-quantization Transformation** reshapes tensor properties before quantization to make them more amenable to low-bit formats. Early methods like SmoothQuant (Xiao et al., 2024) and AWQ (Lin et al., 2024) introduce per-channel scaling factors, either to migrate activation difficulty to the more well-behaved weights or to scale activations to protect the most salient weight channels. More recent works have employed more complex transformations (Shao et al., 2024; Ma et al., 2024). For instance, FlatQuant (Sun et al., 2024) employs a learnable affine transformation aimed at reshaping the data distribution into a flatter, more uniform shape, thereby simplifying the subsequent quantization step. While this data-driven approach offers flexibility, it requires a learning phase and introduces the runtime overhead of the affine transformation during inference. Representing another line of work, QuaRot (Ashkboos et al., 2024) applies a fixed, parameter-free Hadamard transform. This orthogonal transformation deterministically rotates the activations to disperse outliers. Although it avoids a costly learning process, the transform itself still adds computational overhead, and its fixed nature may not be optimal for all data distributions.

In sharp contrast, AURA adopts a fundamentally different post-processing philosophy. AURA does not pre-emptively alter the input data with any transformation—neither learnable nor fixed. Instead, it accepts the original distribution and focuses on compensating for the errors introduced by the quantizer itself. It identifies critical quantization errors and corrects for them by constructing an augmented matrix, fusing the main computation and error compensation into a single standard GEMM operation. This design avoids the runtime transformation overhead inherent to pre-processing methods and offers a more direct, surgical approach to error correction that synergizes well with modern data formats.

**Block-scaling formats** represent a significant advancement in fine-grained quantization, receiving native Tensor Core instruction support in recent architectures like NVIDIA's Blackwell. This hardware-level acceleration effectively resolves the dequantization bottleneck often encountered in

older fine-grained quantization schemes (such as the one used in Atom), where the expansion from low-bit formats relies on less efficient CUDA cores (Lin et al., 2025). AURA is designed to harness the power of these modern formats. It leverages the strong representational capacity of NVFP4 and MXFP4 to store both the primary quantized values and their error compensation terms in a unified low-bit representation. This approach not only fully exploits the performance of hardware-native block-scaling but also maintains compatibility with traditional software-defined schemes like group-wise INT4, demonstrating the versatility of our framework.

## B    DECOMPOSITION OF THE QUANTIZATION ERROR

A linear layer in a transformer performs the operation $\boldsymbol{Y} = \boldsymbol{X}\boldsymbol{W}^T$, where $\boldsymbol{X} \in \mathbb{R}^{N \times C_{in}}$ is the input activation tensor and $\boldsymbol{W} \in \mathbb{R}^{C_{out} \times C_{in}}$ is the weight tensor. Affine quantization maps a high-precision tensor $\mathsf{T}$ (e.g. FP16) to a low-precision format $Q(\mathsf{T})$ using a scale factor $s$. The quantization and dequantization process can be expressed as:

$$Q(\mathsf{T}) = s \cdot \text{round}(\text{clamp}(\frac{\mathsf{T}}{s})) \tag{7}$$

The quantization error for any given tensor $\mathsf{T}$ is then defined as the difference between the original and the dequantized tensor:

$$\mathsf{E}_T = \mathsf{T} - Q(\mathsf{T}) \tag{8}$$

When both activations and weights are quantized, the output of the linear layer becomes $\boldsymbol{Y}_q = Q(\boldsymbol{X})Q(\boldsymbol{W})^T$. The total output error, $\boldsymbol{E}_Y$, is the difference between the ideal output and the quantized output:

$$\boldsymbol{E}_Y = \boldsymbol{Y} - \boldsymbol{Y}_q = \boldsymbol{X}\boldsymbol{W}^T - Q(\boldsymbol{X})Q(\boldsymbol{W})^T \tag{9}$$

Using the error definition from Equation 9, we can substitute $\boldsymbol{X} = Q(\boldsymbol{X}) + \boldsymbol{E}_X$ and $\boldsymbol{W} = Q(\boldsymbol{W}) + \boldsymbol{E}_W$ into Equation:

$$\boldsymbol{E}_Y = (Q(\boldsymbol{X}) + \boldsymbol{E}_X)(Q(\boldsymbol{W}) + \boldsymbol{E}_W)^T - Q(\boldsymbol{X})Q(\boldsymbol{W})^T \tag{10}$$

By expanding the product, we can decompose the total error into three constituent components:

$$\boldsymbol{E}_Y = \underbrace{\boldsymbol{E}_X Q(\boldsymbol{W})^T}_{\text{Term A: Activation Error}} + \underbrace{Q(\boldsymbol{X})\boldsymbol{E}_W^T}_{\text{Term B: Weight Error}} + \underbrace{\boldsymbol{E}_X \boldsymbol{E}_W^T}_{\text{Term C: Second-Order Error}} \tag{11}$$

This decomposition is crucial as it allows us to analyze the contribution of activation and weight quantization errors independently.

## C    FINE-GRAINED QUANTIZATION AND BLOCK-SCALED FORMATS

Symmetric group-wise quantization partitions the $C_{in}$ channels of a tensor $\mathsf{T}$ into groups of size $g$. For each group $i$, a single scaling factor $s_i$ is computed, typically as $s_i = \max(|\mathsf{T}_{j \in i}|)/Q_{max}$, where $Q_{max}$ is the maximum representable value of the low-bit integer type (e.g. 7 for INT4). This strategy effectively isolates the impact of outlier channels, allowing a large scaling factor to be used for a group containing an outlier without degrading the precision of other, well-behaved groups (Heo et al., 2023). The quantization and dequantization are then performed as:

$$Q(\mathsf{T})[*, j] = s_i \cdot \text{round}(\text{clamp}(\frac{\mathsf{T}[*, j]}{s_i})), \quad \forall j \in \text{group}_i \tag{12}$$

where clamp bounds the values within $[-Q_{max}, Q_{max}]$.

Recent GPU architectures have introduced native support for block-scaling formats, which institutionalize the group-wise principle for floating-point types.

**NVFP4:** This format represents a group of 16 floating-point values (using a 2-bit exponent and 1-bit mantissa, E2M1) that share a single, higher precision scaling factor (using a 4-bit exponent and 3-bit mantissa, UE4M3). This effectively implements symmetric group-wise quantization with a fixed group size $g = 16$ (NVIDIA Corporation, 2024a).

**MXFP Formats :** Similarly, these formats represent a block of values (typically 32) using a shared exponent, which acts as a common scaling factor for the mantissas within the block. This is another form of block-scaling that allows the dynamic range of the quantizer to adapt at a fine-grained, hardware-accelerated level (Darvish Rouhani et al., 2023).

## D    ABLATION STUDIES

### D.1    COMPENSATION RATIO

The choice of the compensation ratio k is primarily driven by the desired accuracy, but the distinct outlier-driven nature of activation distributions suggests that significant accuracy gains can be achieved by compensating only a few critical channels. To investigate this, we evaluate the impact of varying the percentage of compensated channels on both 0-shot average accuracy and WikiText2 (Merity et al., 2016) perplexity across Llama and Qwen models. As shown in Figure 8, the results reveal a clear trend of diminishing returns. At very small compensation ratios, the 0-shot average accuracy increases sharply while the perplexity drops significantly. As k continues to increase, the accuracy improvement becomes marginal, even exhibiting slight fluctuations, before eventually resuming a slow upward trend at very high ratios. This confirms that a small number of critical channels are responsible for the majority of the quantization error. For the NVFP4 format, we identify a "sweet spot" in the 4-6% range. This ratio provides the best trade-off, achieving near-maximal accuracy gains while minimizing the computational and memory overhead introduced by the augmented matrix.

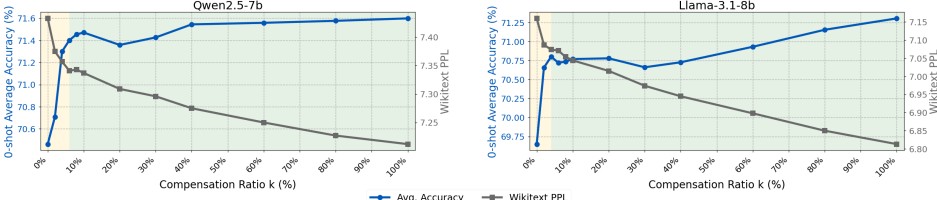

Figure 8: Hyperparameter

### D.2    FUSED KERNELS

AURA's augmented matrix design unifies the main computation and error compensation into a single GEMM call. We benchmark this against a common alternative where the compensation term is added via a fused epilogue. As shown in Figure 9, our single-kernel approach is consistently faster across various matrix scales. It eliminates the overhead of a second kernel launch and avoids a separate memory pass for error accumulation, with the benefits being most pronounced at smaller problem sizes. This validates the efficiency of our data-level fusion strategy. Additionally, AURA fuses pre-processing steps like RMSNorm and channel reordering into its quantization kernel to further minimize overhead.

The results in Figure 10 validate the efficiency of this unified single-kernel approach. Despite a marginal overhead from the augmented dimension, the AURA GEMM is considerably faster than higher-precision formats and outperforms the TensorRT-LLM FP8 baseline, demonstrating a strong balance of speed, simplicity, and adaptability.

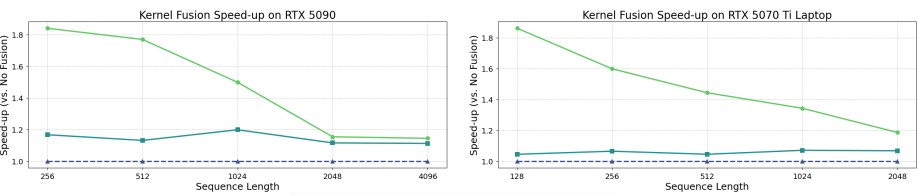

Figure 9: GEMM kernel fusion

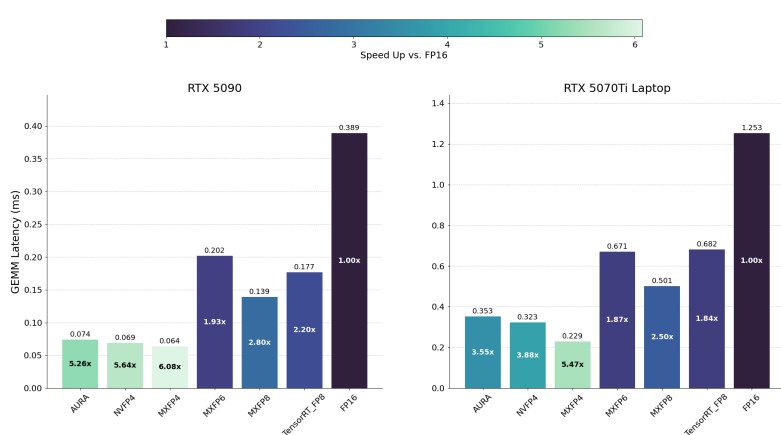

Figure 10: GEMM latency comparison for a 2048x4096x4096 operation on different GPUs. The AURA GEMM, built with CUTLASS, demonstrates a significant performance advantage over the TensorRT-FP8 baseline. MXFP and NVFP GEMMs are built with CUTLASS. The poor performance of MXFP6 is attributed to its limited optimization in current libraries and inherent hardware-unfriendliness, underscoring the risks of non-standard formats.

### D.3 PERFORMANCE IMPACT OF COMPENSATION RATIO

To quantify the performance trade-offs of our compensation mechanism, we ablated the compensation ratio k for the Llama2-7B model. The results in Figure 11 reveal a clear sub-linear relationship between the compensation ratio and actual performance cost. While doubling the GEMM workload (100%) by augmenting the matrix, the end-to-end prefill latency increases by only 36%. The observed non-monotonic fluctuations are characteristic of GEMM execution on modern GPUs, where performance is highly sensitive to matrix dimensions that align with hardware tiling patterns. Nevertheless, the overarching conclusion is that AURA achieves substantial accuracy improvements at a marginal and sub-linear performance cost, underscoring the efficiency of our augmented matrix approach.

### D.4 KV CACHE QUANTIZATION

To evaluate AURA under higher compression, we further applied INT4 group-wise quantization (group size 64, 128) to the KV cache. The results for Llama2-7B and Llama2-13B models are presented in Table 5. As shown, applying KV cache quantization (KV4) leads to an additional accuracy drop of approximately 1.0% for the 7B model and 0.6% for the 13B model compared to AURA with full precision cache. This experiment quantifies the trade-off when both weight and KV cache quantization are applied for deployment in highly resource-constrained environments.

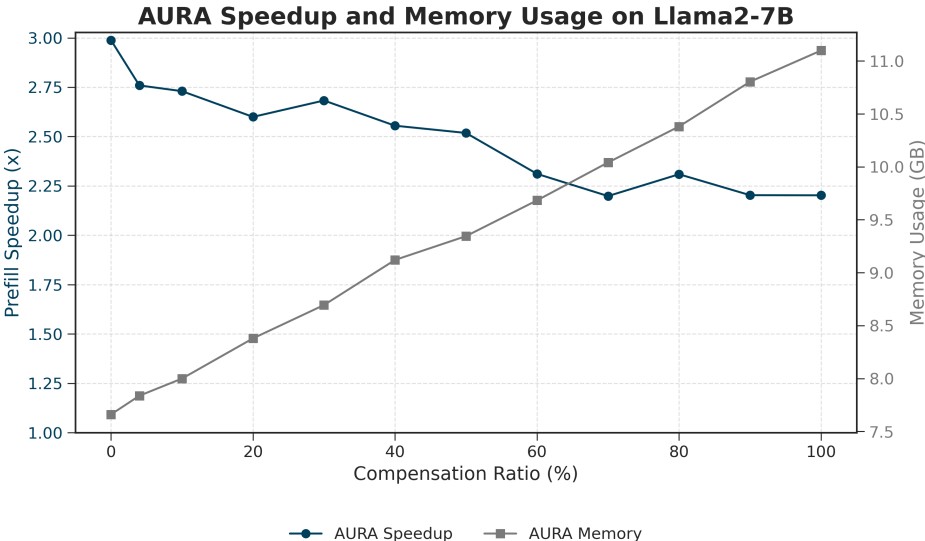

Figure 11: The end-to-end prefill latency was measured on the Llama 2-7B model under a fixed batch size of 8 and a sequence length of 2048, for a range of compensation ratios.

Table 5: The impact of 4-bit KV cache quantization on model performance. We compare the FP16 baseline with AURA and with additional KV cache quantization (KV4). We report 0-shot accuracy (↑) and WikiText2 perplexity (PPL) (↓).

| Model | Method | 0-shot Accuracy | | | | | | | PPL |
|-------|--------|-------|-------|-------|-------|-------|-------|-------|-----|
| | | ARC-C | ARC-E | BQ | HG | PQ | WG | Avg. | |
| Llama2-7B | FP16 | 46.25 | 74.58 | 77.74 | 75.92 | 79.00 | 68.98 | 70.41 | 5.47 |
| | KV16 | 44.03 | 71.72 | 76.27 | 74.84 | 78.18 | 68.27 | 68.89 | 5.78 |
| | KV4-64 | 44.37 | 71.68 | 75.66 | 74.34 | 78.36 | 67.72 | 68.69 | 5.85 |
| | KV4-128 | 42.06 | 71.46 | 74.83 | 73.99 | 78.45 | 66.61 | 67.90 | 5.93 |
| Llama2-13B | FP16 | 48.98 | 77.48 | 80.76 | 79.35 | 80.58 | 71.98 | 73.19 | 4.89 |
| | KV16 | 49.15 | 76.22 | 80.18 | 77.99 | 79.54 | 71.59 | 72.45 | 5.10 |
| | KV4-64 | 48.21 | 76.30 | 80.15 | 77.78 | 79.87 | 71.35 | 72.28 | 5.15 |
| | KV4-128 | 46.93 | 75.55 | 78.99 | 77.37 | 79.87 | 72.53 | 71.87 | 5.20 |

## D.5 DATA FORMATS

To validate its versatility, we integrated the AURA framework with three distinct quantization schemes: NVFP4, MXFP4, and a group-wise INT4. As shown in Table 6, AURA consistently and substantially improves the performance of every underlying format. The optimal compensation ratio k is highly format-dependent (6% for NVFP4, 12% for MXFP4, 24% for INT4), reflecting the interplay between a format's raw representational power and AURA's ability to compensate for its specific weaknesses. As emerging data formats continue to evolve with better representational capabilities, we anticipate that AURA will require an even smaller k to achieve near-lossless performance, confirming its status as a robust and future-proof solution.

## D.6 BENCHMARKS ON MATH

To evaluate AURA's ability to preserve specialized reasoning capabilities, we benchmarked it on the Qwen-Math models, with results shown in Table 7, including GSM8K (Cobbe et al., 2021), MMLU-STEM (Hendrycks et al., 2021), CMATH (Wei et al., 2023), and CN Middle School Math 24. The

Table 6: Demonstration of AURA's generality on Qwen2.5-7B. AURA is applied as a plug-in enhancement to three different 4-bit base formats: NVFP4, MXFP4, and standard INT4. We report 0-shot accuracy (↑) and WikiText2 perplexity (PPL) (↓). The best result among all quantization methods is in **bold**.

| Method | 0-shot Accuracy | | | | | | | PPL |
|---|---|---|---|---|---|---|---|---|
| | ARC-C | ARC-E | BQ | HG | PQ | WG | Avg. | |
| FP16 | 51.28 | 77.53 | 84.68 | 78.98 | 79.71 | 73.24 | 74.24 | 6.85 |
| NVFP4 | 50.77 | 78.70 | 82.84 | 77.29 | 78.73 | 69.14 | 72.91 | 7.43 |
| **AURA-NVFP4** | **51.02** | **78.70** | **83.91** | **77.45** | **79.87** | **71.43** | **73.73** | **7.34** |
| MXFP4 | **50.94** | 75.38 | 82.48 | 75.18 | 77.37 | 66.93 | 71.38 | 8.21 |
| **AURA-MXFP4** | 50.26 | **76.47** | **83.36** | **76.33** | **79.11** | **68.90** | **72.40** | **7.73** |
| INT4 | 43.26 | 68.35 | 76.73 | 71.59 | 76.39 | 63.06 | 66.56 | 9.72 |
| **AURA-INT4** | **47.18** | **73.65** | **79.82** | **75.67** | **78.35** | **67.56** | **70.37** | **8.19** |

Table 7: Performance on mathematical reasoning benchmarks for the Qwen2.5-Math-7B-Instruct model. We report pass@1 accuracy. MMLU-S refers to MMLU STEM, and CN Middle refers to the Chinese Middle School Math benchmark.

| Method | GSM8K | MMLU-S | CMATH | CN Middle |
|---|---|---|---|---|
| FP16 | 95.8 | 77.8 | 91.2 | 72.3 |
| AURA | 95.2 | 70.2 | 92.0 | 76.2 |
| FP8 | 95.5 | 68.7 | 91.7 | 74.3 |

findings demonstrate that AURA achieves performance highly comparable to the FP16 baseline. On Qwen-Math, AURA maintains strong performance, even outperforming the FP16 baseline on two of the four benchmarks. Furthermore, AURA shows a clear advantage over other quantization baselines, surpassing FP8 on the MMLU-STEM task and outperforming Atom across all coding benchmarks.

# E    THE USE OF LARGE LANGUAGE MODELS (LLMS)

We utilized large language models (LLMs) for grammatical correction and stylistic improvements. The research design, experiments, analysis, and conclusions were developed entirely by the authors.

