# OpenReview forum: "AURA: Augmented Representation for Unified Accuracy-aware Quantization"
_ICLR.cc/2026/Conference — ICLR 2026 Conference Withdrawn Submission_

### Official Review · Reviewer_gXX2 · 2025-10-29

**Soundness:** 2
**Presentation:** 2
**Contribution:** 2
**Rating:** 2
**Confidence:** 5

**Summary:**

This paper presents AURA, a framework for efficient low-bit quantization of Large Language Models (LLMs). AURA proposes a theoretically motivated, accuracy-aware metric for identifying critical channels and compensating their quantization errors through a unified, low-bit augmented matrix formulation. By decoupling error compensation from GEMM kernels, AURA aims to achieve state-of-the-art accuracy and hardware compatibility without the runtime overhead of prior transformation-based or mixed-precision methods. Experiments across Llama and Qwen models show improved performance on a range of benchmarks and claim substantial inference-speed and memory benefits.

**Strengths:**

A substantive assessment of the strengths of the paper, touching on each of the following dimensions: originality, quality, clarity, and significance. We encourage reviewers to be broad in their definitions of originality and significance. For example, originality may arise from a new definition or problem formulation, creative combinations of existing ideas, application to a new domain, or removing limitations from prior results.
● The authors develop a clear, mathematically grounded channel ranking for accuracy-aware compensation. This addresses well-known limitations in prior heuristics by explicitly quantifying channel-wise error impact.
● The use of fused CUDA kernels and empirical benchmarks shows notable advancements: Figure 6 demonstrates substantial improvements in prefill latency and memory usage, and Figure 10 compares kernel throughput to baselines effectively.

**Weaknesses:**

A substantive assessment of the weaknesses of the paper. Focus on constructive and actionable insights on how the work could improve towards its stated goals. Be specific, avoid generic remarks. For example, if you believe the contribution lacks novelty, provide references and an explanation as evidence; if you believe experiments are insufficient, explain why and exactly what is missing, etc.

● The main results (Tables 2, 3, and 6) focus primarily on Llama and Qwen variants. There is minimal direct comparison with very recent and highly relevant quantization techniques such as QTIP [1], FlashAttention-3 [2], Delta-CoMe [3], GPTAQ [4], GuidedQuant [5], OSTQuant [6]. These omissions weaken claims of state-of-the-art generality and may mask important trade-offs in accuracy or hardware compatibility in broader contexts.
● The central theoretical contribution is the Frobenius-norm-driven sensitivity ranking. However, the subsequent decision to concatenate [quantized activation | critical channel | quantized error] is treated primarily as an engineering mechanism, not developed from first theoretical principles. There’s little formal analysis or theoretical justification for the specific form of the augmented matrix or an explicit characterization of its error bounds. For instance, how does the aggregated quantization error in the fused matrix propagate downstream? Is there a risk of introducing new forms of correlated noise, or are there worst-case guarantees for error compensation in adversarial activation scenarios? A more rigorous mathematical treatment would improve confidence in the method’s generality.
● While Figure 5 (selection strategies) and Figure 8 (compensation ratio) validate the metric and the trade-off for Llama and Qwen, there is no analysis on domain adaptation (e.g., what happens if the calibration set is misaligned with deployment data, beyond the three chosen datasets in Table 4?). This is particularly pertinent since LLM deployment often encounters data outside the calibration domain. Additionally, additional settings (other architectures, data types, training regimes) could reveal hidden limitations of the approach.
● The framework appears tailored to Transformer-based architectures and block-scaling formats with specific hardware features (e.g., as detailed in the NVFP4/MXFP discussion and Figure 10). There is little to no discussion on limitations (hardware requirements, compatibility with novel or hybrid models, scalability to sub-4-bit formats as briefly mentioned in the conclusion). Where does the approach break down or become suboptimal?
● None of the main empirical tables (e.g., Table 2, Table 6) report variance or statistical significance for the results, which is concerning given the often small absolute accuracy gains (especially with a compensation ratio as low as 6%). Are these differences reproducible across calibration sets and seeds? Without these details, much of the "state-of-the-art" claim is vulnerable.
● The use of WikiText2 and a relatively narrow set of calibration/evaluation datasets, even with some variety (Table 4), raises questions about generalization. Although Table 7 includes some math benchmarks, the breadth of quantitative evidence is skewed toward standard tasks.

[1] Tseng, Albert, et al. "Qtip: Quantization with trellises and incoherence processing." Advances in Neural Information Processing Systems 37 (2024): 59597-59620.
[2] Shah, Jay, et al. "Flashattention-3: Fast and accurate attention with asynchrony and low-precision." Advances in Neural Information Processing Systems 37 (2024): 68658-68685.
[3] Ping, Bowen, et al. "Delta-come: Training-free delta-compression with mixed-precision for large language models." Advances in Neural Information Processing Systems 37 (2024): 31056-31077.
[4] Li, Yuhang, et al. "GPTAQ: Efficient Finetuning-Free Quantization for Asymmetric Calibration." Forty-second International Conference on Machine Learning.
[5] Kim, Jinuk, et al. "GuidedQuant: Large Language Model Quantization via Exploiting End Loss Guidance." Forty-second International Conference on Machine Learning.
[6] Hu, Xing, et al. "OSTQuant: Refining Large Language Model Quantization with Orthogonal and Scaling Transformations for Better Distribution Fitting." The Thirteenth International Conference on Learning Representations.

**Questions:**

Please list up and carefully describe any questions and suggestions for the authors. Think of the things where a response from the author can change your opinion, clarify a confusion or address a limitation. This is important for a productive rebuttal and discussion phase with the authors.
Please address my concerns in Weaknesses.

---

### Official Review · Reviewer_kgkK · 2025-10-29

**Soundness:** 3
**Presentation:** 2
**Contribution:** 2
**Rating:** 4
**Confidence:** 4

**Summary:**

This paper introduces a quantization framework designed to mitigate accuracy degradation in quantized models. The core idea is to employ a metric for identifying the most important channels and activations, and then to strategically compensate for quantization error by augmenting this critical information back into the quantized weights and activations. This approach aims to preserve key information more effectively than standard methods. Empirical results demonstrate that the framework outperforms baselines. However, the paper would be strengthened by providing more detailed explanations of the methodology, and a discussion regarding the final compression ratio achieved is needed to fully assess the practical utility of the approach. Addressing these points would improve the paper's clarity and quality.

**Strengths:**

This paper presents a well-motivated quantization framework with a clear and streamlined concept: augmenting quantized weights and activations by strategically reinjecting a portion of the quantization error to enhance accuracy. The methodology is intuitively appealing. The authors provide evidence for its effectiveness, demonstrating superior performance over compared baseline methods. The approach also delivers practical value by achieving better end-to-end inference efficiency compared to the FP8 baseline.

**Weaknesses:**

The paper's core contributions are undermined by a lack of rigorous justification for its foundational premises and key methodological choices. The central claim that activation error is the primary source of quantization degradation is supported only by intuitive observations that contradict established literature on weight distributions, and the mathematical argument (based on an upper-bound analysis) used to derive the importance metric is logically flawed. Furthermore, the novelty of the proposed metric is unclear, as it closely resembles existing techniques from the pruning literature. The practical applicability is limited by the use of a fixed, seemingly arbitrary compensation ratio (4-6%) without validation of its generalizability or an automated selection mechanism. These issues, combined with the omission of standard metrics like equivalent bit-width for comparison, make it difficult to assess the true effectiveness and novelty of the proposed framework.

**Questions:**

1. The paper's foundational motivation rests on the claim that activation error is the primary source of accuracy degradation in LLM quantization, supported by an intuitive inference in Lines 109-112. To strengthen this core premise, could you provide more rigorous evidence? For instance, a theoretical analysis of error propagation or controlled empirical results that disentangle and quantify the individual impact of weight quantization error versus activation quantization error would make this claim significantly more compelling.

2. The explanation in Lines 113-115, which attributes the primary source of error to activations (due to *"well-distributed weights"* and *"outlier-equipped activations"*), lacks rigor and may oversimplify the reality of quantizing LLMs. This characterization appears to contradict established observations in the literature [1, 2], where weight distributions are also known to be long-tailed and can contain significant outliers.

3. Equation (4) deduces an upper bound for the total error, leading to the claim that *"the total error is dominated by channels with the highest contribution norms."* This reasoning is not rigorous. An upper bound describes a worst-case scenario; it cannot be used to conclusively analyze what typically or primarily contributes to the error in practice, as the equality condition may never be achieved. To strengthen this argument, the analysis should be based on a ​​lower bound​​ or, more appropriately, an ​​expected value​​ or ​​average-case​​ analysis of the error.

4. Equation (6) introduces a metric to identify the importance of channels for error compensation. However, a very similar formulation, often based on the product of weight norms and activation magnitudes, is commonly used in the pruning literature (e.g., in works like [3]) to identify salient weights or channels. Could you please clarify the specific novelty of their proposed metric in this context?

5. The paper uses a compensation ratio of 4-6%. How can we guarantee that this specific ratio is generally effective across diverse model architectures, scales, and tasks? Is there empirical or theoretical evidence to support that this narrow range is a universal constant, or could it be highly model-dependent?

6. For real-world usability, a fixed, pre-defined compensation ratio is suboptimal. Could there be a simple, practical, and automated mechanism to determine this ratio dynamically for a specific target model?

7. To enable a more precise and fair comparison of the compression efficiency achieved by different methods, could you please add a column in Table 2 indicating the equivalent bit-width after quantization for each method?


[1] Kim, S., Hooper, C., Gholami, A., Dong, Z., Li, X., Shen, S., ... & Keutzer, K. (2023). Squeezellm: Dense-and-sparse quantization. arXiv preprint arXiv:2306.07629.

[2] Zhao, P., & Yuan, X. (2025). GANQ: GPU-Adaptive Non-Uniform Quantization for Large Language Models. arXiv preprint arXiv:2501.12956.

[3] Sun, M., Liu, Z., Bair, A., & Kolter, J. Z. (2023). A simple and effective pruning approach for large language models. arXiv preprint arXiv:2306.11695.

---

### Official Review · Reviewer_Fsxs · 2025-10-30

**Soundness:** 2
**Presentation:** 3
**Contribution:** 3
**Rating:** 4
**Confidence:** 4

**Summary:**

This paper introduces AURA, a novel quantization framework that resolves the trade-off between slow, transformation-based methods and hardware-specific, mixed-precision schemes. AURA proposes a metric to identify critical channels and compensates for their quantization errors by constructing a low-bit augmented matrix. This fuses the main computation and error correction into a single standard GEMM operation, decoupling the algorithm from the hardware kernel and achieving good accuracy and system performance.

**Strengths:**

1.	The idea of using an augmented matrix is a clever and practical approach. By expanding the K-dimension of the GEMM to include error compensation terms, AURA effectively decouples the complex quantization logic from the highly-optimized GEMM kernel. This is an engineering advantage over mixed-precision methods that require custom kernels, making the solution more portable.

2.	The framework's design demonstrates a strong focus on practical, system-level efficiency. The authors present a high-performance implementation where complex pre-processing steps like RMSNorm, channel reordering, and error quantization are fused into a single optimized kernel, minimizing latency and memory I/O overhead. This practical engineering is complemented by the method's proven efficiency. As shown in Table 6, AURA functions as a plug-in enhancement that consistently improves performance across multiple underlying 4-bit formats (NVFP4, MXFP4, INT4). This adaptability makes the framework robust against the evolution of hardware and data formats.

**Weaknesses:**

1. The paper demonstrates robustness to different calibration datasets (Table 4). However, it lacks a crucial analysis of the method's sensitivity to the calibration process itself. Key questions remain unadressed:
a) How sensitive is AURA to the different random seeds in selecting the calibration samples? Will the ordering pattern and the set of "critical channels" be stable across different runs? A high variance would question the robustness of the proposed metric in practice.
b) How sensitive is AURA to the number of calibration samples? The paper uses 32 samples, but there is no justification or ablation provided for this choice. An analysis showing the trade-off between the number of samples and final model accuracy would be necessary to understand this dependency.

2.  Lack of Direct Overhead Comparison with Baselines: AURA achieves its gains by introducing a computational and memory overhead, specifically by augmenting the K-dimension of the GEMM by 4-6%. While the authors analyze the performance impact of this overhead (Figure 6), the comparison to baselines feels incomplete from an overhead perspective. Methods like FlatQuant also introduce extra parameters (for learnable transformations), and methods like QuaRot require a parameter-free but computationally expensive rotation. The paper should provide a more direct comparison of these different overheads. For a truly fair comparison, a table is needed, detailing the extra memory (for parameters or augmented activations) and estimated extra FLOPs for each method (e.g., AURA's augmented GEMM vs. FlatQuant's pre-transformation matrix multiplication) as well as end-to-end performances. Without this, it is difficult to assess whether AURA's accuracy-performance trade-off is genuinely superior or if it simply operates at a different point on the overhead spectrum.

**Questions:**

See the weakness part.

---

### Official Review · Reviewer_v2BP · 2025-10-31

**Soundness:** 3
**Presentation:** 3
**Contribution:** 2
**Rating:** 2
**Confidence:** 4

**Summary:**

This paper proposes AURA to compensate for outliers present in activation tensors. The proposed solution identifies the most sensitive channels in the activations, then corrects their quantization error during the GEMM computation. The entire tensor, including the quantization error, is still homogeneously quantized so that there is no extra overhead to be paid during GEMM computation aside from the extra compute. The first main contribution is an offline calibration-based algorithm for identifying the most sensitive channels in the input of each linear layer. This allows for fast computation during inference, since the most expensive operations entail the computation of the quantization error of a subset of channels of the activation tensors, and the permute operations of the activation tensor itself for each linear layer. The authors also then propose efficient fused kernels to reduce the overhead introduced by these new operations.

**Strengths:**

1. The paper is well-written and easy to follow, with a clearly motivate problem (addressing the accuracy drop that comes from aggressively quantizing sensitive channels).
2.  The core idea of appending and quantizing the error associated with critical channels is neat and avoids modifying GEMM kernels.
3. The proposed method is simple, easy to implement, and computationally efficient.
4. The authors propose open-source accelerated kernels to minimize the overhead of the error correction process.

**Weaknesses:**

1. Table 2 is confusing. NVFP4 is listed as a “method,” but it is actually a datatype. Since other algorithms in the table (e.g., QuaRot, FlatQuant) can also be applied to the NVFP4 format, the comparison is unclear. As written, the results suggest that AURA only marginally outperforms simple rounding when quantizing to NVFP4.
2. It is unclear how results for other SOTA methods were obtained (re-implementation, external libraries, or prior reports) and if all algorithms are compared with the same datatype; this needs clarification to ensure fair comparison.
3. The reported results are nearly identical and it’s unclear if the ranking of methods is stable. Either run-to-run variation or additional models of varied sizes and architectures would be sufficient to address this.
4. The memory overhead introduced by offline expansion is not clearly characterized, especially for MXFP4 and INT4 where the ratio of sensitive channels seems high.
5. The performance analysis emphasizes FP8 comparisons, with only limited NVFP4 results in the Appendix. Since NVFP4 achieves strong accuracy, the paper should present full NVFP4 results in the main text to enable a fair and consistent evaluation.
6. The literature review is incomplete. It focuses primarily on mixed-precision methods, quantization transformations, and block-scaled formats, but omits prior work on model expansion [1–4], which is the line of research most directly relevant to this study. While each of these works differs from the current proposal, the related work section would be incomplete without acknowledging at least some of them (and/or potentially others).
7. The paper’s discussion of transformation-based methods is somewhat unfair. It states (already in the abstract and around line 53) that such methods “generate” significant overhead, but several existing works show that the overhead can be either minor in practice or completely merged away (see R1 and R2 in SpinQuant). Such claims should be either significantly toned down or supported.
8. The introduction claims that methods such as GPTQ leave activations unquantized (around line 40), but there is nothing in GPTQ that prevents activation quantization. Indeed, W4A8-level results exist in the literature, including in GPTQ-style pipelines. This is again an unfair treatment of existing works.
9. The notation is loose in places (see questions).


References:

1. Zhao et al. (2019), Improving neural network quantization without retraining using outlier channel splitting.

2. Adepu et al. (2024), Framequant: Flexible low-bit quantization for transformers.

3. Li et al (2024), Svdquant: Absorbing outliers by low-rank components for 4-bit diffusion models

4. Franco et al. (2025), Improving quantization with post-training model expansion.

**Questions:**

1. I think this is important. The notation around the quantizers is too loose. In Eq.  (Y_q = Q(X) Q(W)^T, line 104) the paper should make it explicit that the two Q’s need not be the same. In practice, they usually are not, and even different scaling factors imply different Q’s. This becomes even more important in lines 147–148, where it looks like a single symbol Q is standing for (at least) three different quantizers: for the weights, for X_n / X_c, and for E_c.
2. The sentence around line 134 (“it is theoretically incomplete…”) is vague at the point it appears; it seems to foreshadow Eqs. (4)–(6), but those have not been introduced yet.
3. The bound in Eq. (4) is only an upper bound and may be quite loose. While empirically it seems to lead to good results, is there a theoretical reason to believe that minimizing the upper bound leads to minimizing the object of interest?
4. “Dynamic compensation” in the conclusion is a bit vague.
5. Appendix B is longer than it needs to be, the derivation can be done in one line and moved to the main text.
6. Do you have any deeper intuition as to why the compensation ratio should change for different models? It would be useful to know what compensation ratios are good for what models and why. For example, it would be interesting to see if there is cross-model variability for the compensation ratio by testing more models within the Qwen and Llama families.
7. Considering the higher compensation ratio required for MXFP4, it would be good to have some numbers in terms of overhead added for this specific data type versus baseline MXFP4. Have you run such experiments?

---

### Note · Authors · 2025-11-28

**Comment:**

We sincerely thank all four reviewers for their thoughtful and detailed evaluations. We appreciate that the reviewers carefully assessed our paper and provided valuable feedback regarding both its strengths and areas for improvement. After thorough consideration, we have decided to withdraw our submission to extensively revise the work based on these constructive comments. Finally, we would like to express our gratitude to ICLR 2026 for the opportunity to receive such high-quality feedback, which will greatly benefit the future development of our research.

**Withdrawal Confirmation:**

I have read and agree with the venue's withdrawal policy on behalf of myself and my co-authors.